**Protocol**

# Repetitive transcranial magnetic stimulation for depression after basal ganglia ischaemic stroke: protocol for a multicentre randomised double-blind placebo-controlled trial

Ying Tang,[1] Aimin Chen,[2] Shuzhen Zhu,[2] Li Yang,[2] Jiyuan Zhou,[3] Suyue Pan,[4] Min Shao,[5] Lianxu Zhao[2]

[1]Zhujiang Hospital of Southern Medical University, Guangzhou, China
[2]Department of Neurology, Zhujiang Hospital of Southern Medical University, Guangzhou, China
[3]School of Public Health, Southern Medical University, Guangzhou, China
[4]Department of Neurology, Nanfang Hospital of Southern Medical University, Guangzhou, China
[5]Department of Neurology, Sichuan Bayi Rehabilitation Center, Chengdu, China

**Correspondence to**
Dr Lianxu Zhao MD and PhD;
zhaolianxu@smu.edu.cn

## ABSTRACT

**Introduction** Studies suggest that repetitive transcranial magnetic stimulation (rTMS) is effective for the treatment of depression and promotes the repair of white matter. This study aims to assess the effectiveness of rTMS in treating depression after basal ganglia ischaemic stroke and to examine whether such effects are related to restoration of white matter integrity.

**Methods and analysis** Sixty-six participants will be recruited from Zhujiang Hospital, Nanfang Hospital and Sichuan Bayi Rehabilitation Hospital and randomised in a 1:1 ratio to receive active rTMS treatment or sham rTMS treatment in addition to routine supportive treatments. The data will be collected at 0, 2 and 4 weeks after the commencement of treatment. The primary outcome is the measurement of 24-item Hamilton Depression Rating Scale scores, and the secondary outcomes include diffusion tensor imaging results and the results of neuropsychological tests including the National Institutes of Health Stroke Scale, Activities of Daily Living Scale, Montreal Cognitive Assessment, Clinical Global Impressions scales, Aphasia Battery in Chinese, Social Support Revalued Scale and Medical Coping Modes Questionnaire.

**Ethics and dissemination** This study has been approved by the Ethics Committee of Zhujiang Hospital of Southern Medical University. The findings will be disseminated by publication in a peer-reviewed journal and by presentation at international conferences.

**Trial registration number** NCT03159351.

## INTRODUCTION
### Background and rationale

Stroke is a global health problem and, in 2015, the number of deaths related to stroke reached 6.326 million around the world, among which 45.89% were attributed to ischaemic stroke.[1] Depression occurs frequently in stroke survivors and has a significant impact on post-stroke rehabilitation.[2] The conventional treatment of post-stroke depression (PSD) relies on supportive therapy and

### Strengths and limitations of this study

► The trial will focus on the efficacy of repetitive transcranial magnetic stimulation (rTMS) in patients with post-stroke depression (PSD) with the purpose of providing a mature approach to the treatment of PSD.

► It will explore the effect of high-frequency stimulation on the left and low-frequency stimulation on the right dorsolateral prefrontal cortex in patients with PSD, aiming to obtain a higher level of evidence for the stimulation parameter settings in rTMS treatment of patients with PSD.

► Non-invasive diffusion tensor imaging based on objective assessment of nerve fibres will be used to evaluate the therapeutic effect of rTMS on PSD.

► The patients will receive only short-term treatment and will be followed up for 1 month.

► Participants from only two provinces in China will be included.

medications, but a proportion of patients can remain refractory to treatment.[3] Studies have shown that repetitive transcranial magnetic stimulation (rTMS) has a definite antidepressant effect on major depression,[4] and a recent meta-analysis also suggests the beneficial effects of rTMS on PSD, but such studies should be carefully examined for their heterogeneity and potential biases.[5] To our knowledge, no studies have been reported that examine the effect of rTMS on PSD in China, and the mechanism by which rTMS improves PSD remains to be clarified. In addition, the stimulation parameter settings in rTMS treatment still await standardisation, and it is difficult to determine the optimal frequency (between 5 Hz and 20 Hz) in high-frequency rTMS of the left dorsolateral prefrontal cortex (DLPFC).

The basal ganglia area is the predilection site of stroke. This area, which is not an exact anatomical concept but one originating from neuroimaging, consists of the basal ganglia, the capsula interna and the surrounding white matter[6]; in other words, the basal ganglia area covers most of the subcortical structures that constitute the limbic–cortical–striatal–pallidal–thalamic (LCSPT) circuit.[7] The limbic system is known to be closely related with emotional regulation,[8] and as the LCSPT circuit is the most important part of the limbic system, LCSPT dysfunctions are thought to contribute to the occurrence of depression.[9] Investigators reported a high incidence of depression after cerebral infarction in the basal ganglia area.[10] Our group previously found that structural damage of the LCSPT circuit, as a result of white matter damage following cerebral infarction in the basal ganglia area, was associated with a high likelihood of PSD.[11] The nuclei in the LCSPT circuit are connected via nerve fibres, and damage to these nerve fibres disrupts the integrity of the white matter as shown by MRI.[12]

As numerous studies have demonstrated the beneficial effect of rTMS treatment on white matter integrity,[13–15] we hypothesised that the antidepressant effect of rTMS in patients with depression following cerebral infarction in the basal ganglia area is related to its action on the LCSPT circuit. The aim of this multicentre randomised controlled trial is to assess the effectiveness of rTMS in treating PSD and to investigate the role of the LCSPT circuit in the therapeutic effect of rTMS for PSD.

## METHODS
### Study setting
The study will be conducted in two university hospitals, Zhujiang Hospital and Nanfang Hospital affiliated to Southern Medical University (Guangzhou, China) and in Sichuan Provincial Bayi Rehabilitation Hospital, a general hospital in Chengdu, China. The patients will be assessed for enrolment, and eligible patients will be hospitalised for treatment in the departments of neurology in the three hospitals.

### Trial design
This randomised double-blind placebo-controlled multicentre superiority trial is designed to recruit two parallel groups, and randomisation will be performed by block randomisation in a 1:1 allocation ratio. The study design is shown in the flow chart in figure 1.

### Participant enrolment and eligibility criteria
The participants will be recruited by advertisement on the internet and using posters in hospitals. Potential eligible patients will be screened for enrolment according to the inclusion and exclusion criteria listed in table 1.

### Ethical considerations
This study has been approved by the Ethics Committee of Zhujiang Hospital of Southern Medical University.

The methods, purpose and potential risks of this study will be fully explained to the participants and their family members. Each patient will be asked to sign three informed consents before participating in the study.

### Sample size calculation
The sample size was calculated based on the data of Valiengo et al,[16] who reported a response rate of 0.041 in patients receiving sham rTMS treatment and a rate of 0.375 in patients having active rTMS treatment. Considering the low efficiency of sham rTMS treatment, we calculated the sample size using the following formula:

$$n_1 = n_2 = \frac{(Z_{\alpha/2} + Z_\beta)^2}{2(\sin^{-1}\sqrt{P_2} \ \sin^{-1}\sqrt{P_2})^{2\circ}}$$

The result showed that a sample size of 26 in each group was sufficient by assuming a two-tailed significance level of 5% with a statistical power of 90%. We estimate that the dropout rate will be less than 25%, and considering the need for randomisation, we assume that at least 33 participants are required in each group.

### Randomisation and blinding
For randomisation of the patients, random numbers will be generated using SAS9.4 software by statisticians based on a randomised block design with a block length of s. The magnetic stimulation coils for active and sham treatments have identical appearances and will be numbered by a statistician. The random numbers will be printed and sent to each trial centre in sealed envelopes. Upon enrolment of an eligible patient, the investigator opens the envelope to reveal the number of the magnetic stimulation coil to be used, without knowing whether the patient receives active or sham treatment. In this double-blind trial, both the participants and investigators are blinded to the nature of treatment (active or sham).

### Interventions
The participants enrolled in this trial will receive either active or sham rTMS treatment in addition to routine supportive treatments including antiplatelet agents, blood pressure control, neurotrophic therapy, physical therapy and cognitive rehabilitation prescribed based on the condition of the individual participant. The enrolled patients will not receive any other antidepressants, anxiolytics or hypnotic treatments that may have an impact on depression treatment during the study. All the participants will receive psychiatric and neuropsychological tests and MRI examination at the time of enrolment and at the end of treatment. rTMS will be performed using a Mag Pro R30 device (Mag Venture Company, Frum, Denmark) via 121 mm circular coils (MCF-125). The participants will receive active or sham treatment 5

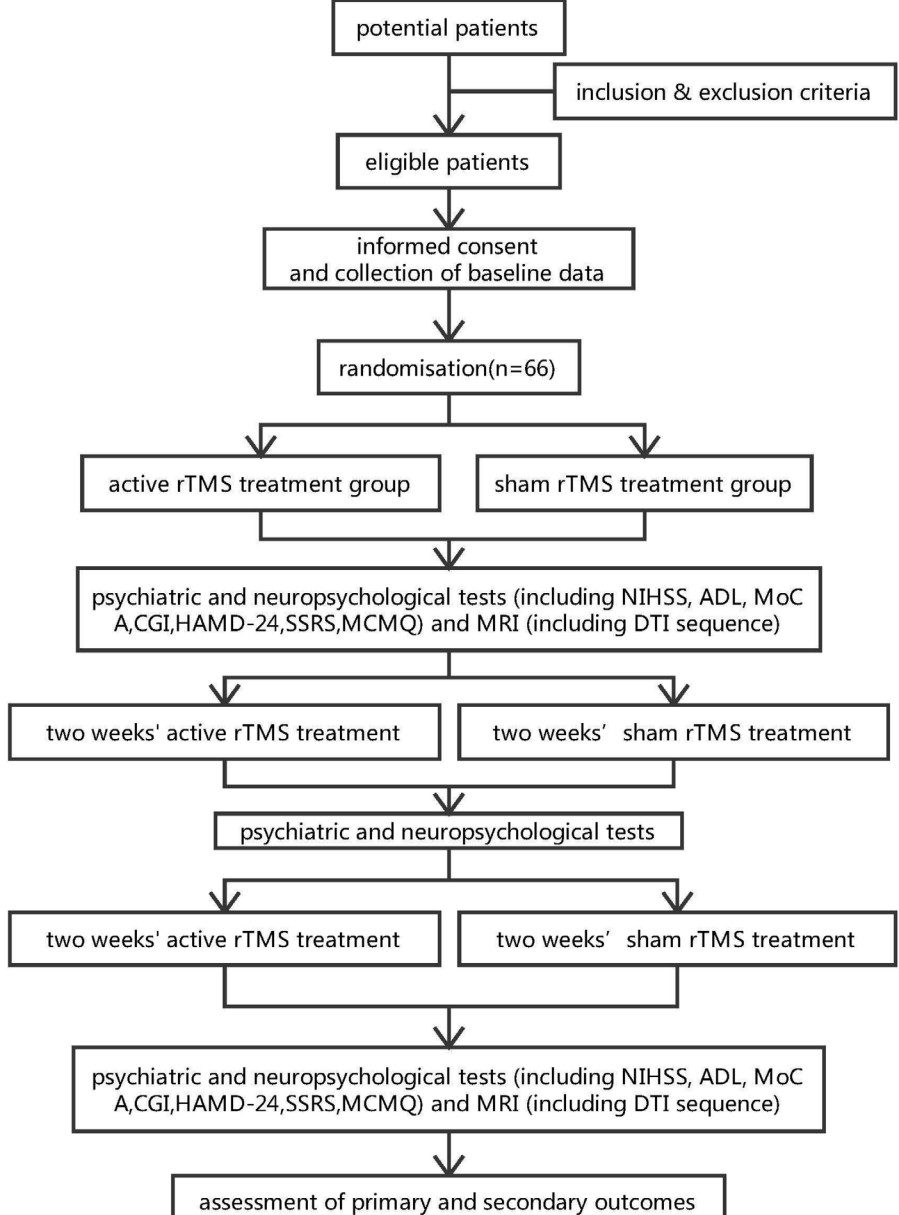

**Figure 1** Flow chart of the trial design.

days per week for four consecutive weeks. The motor threshold (MT) is defined as the lowest stimulation intensity capable of inducing at least 50 µV peak-to-peak amplitude of motor evoked potentials in at least half of 10 trials. The detailed parameters of the protocols of active rTMS treatment are listed in table 2.

### MRI

On enrolment and at the end of the trial, the patient will receive a MRI examination using a 3.0T PHILIPS Magnetic Trio Tim scanner. For the standard investigation, MRI will be performed using the following scan parameters for different sequences: for T1-weighted sequence, time of echo (TE) 20 ms, time of repetition (TR) 2000 ms, number of signal averaged (NSA) 1, number of slices 32, slice thickness 3 mm; for T2-weighted sequence, TE 80 ms, TR 3000 ms, NSA 2,

32 slices, slice thickness 3 mm; for FLAIR sequence, TE 80 ms, TR 3000 ms, NSA 2, 32 slices, slice thickness 3 mm; for diffusion tensor imaging (DTI) using single short echo planar imaging (SE-EPI), TE 68 ms, TR 2300 ms, NSA 1, 32 slices, slice thickness 3 mm, $b_0=0$, $b_1=1000$, matrix=512*512. Once the original data have been obtained, an experienced imaging physician will select the caudate nucleus, lentiform nucleus, dorsal medial nucleus of the thalamus and the nucleus of the thalamus as the regions of interest (ROIs) at the imaging layer of the basal ganglia. The three-dimensional fibre bundle imaging technique will be used to display the fibrous pathway of the dorsal basal ganglia system and the fibrous pathway of the ventral basal ganglia system. The DTI parameters will be obtained, including the fractional anisotropy (FA), apparent

| Table 1 | Inclusion and exclusion criteria |
| --- | --- |
| **Criteria** | **Description** |
| Inclusion criteria | ► First-time ischaemic stroke with clinical and MRI or CT findings of basal ganglia ischaemic stroke and a DSM-IV diagnosis of depression due to stroke (ICD-10-CM code 293.83[F06.32])<br>► Age 25–75 years with a recent (from 3 weeks to 3 months) ischaemic stroke<br>► Clear signs of neurological deficits in the acute phase<br>► Clear consciousness<br>► Right-handedness |
| Exclusion criteria | ► Aphasia or severe cognitive impairment, severe hearing impairment, or severe language comprehension deficits due to other causes<br>► Other cerebral diseases such as Parkinson's disease, encephalitis, dementia, multiple sclerosis, head injury<br>► Severe systemic disease or ongoing neoplasia<br>► Ongoing postoperative recovery<br>► Prior history of depressive disorders or major trauma within 1 year, severe depression or any other severe mental disorders<br>► Current or prior antidepressant use for any reason<br>► Addiction to drugs, alcohol or other substances<br>► Contraindications of MRI scan and rTMS treatment such as pacemaker implantation, a history of epilepsy, major head trauma and seizures<br>► Pregnant or breast-feeding women<br>► Participation in other clinical research projects<br>► Refusal to sign the informed consent of this study |

diffusion coefficient (ADC) and neural fibre number (NFN). The MRI scanners at the three centres will be tested for homogeneity before the trial, while ROI selection and DTI data acquisition will be performed by an experienced radiologist using PHILIP 2.1 software.

### Outcome assessment
#### Baseline assessment
Demographic data will be collected from the patients, including age, gender, height, body weight, education level, occupation and marital status. Vital signs will be measured including body temperature, pulse, respiration and blood pressure. Lesion characteristics including lateralisation and the number and size of the lesions will be recorded. Data including past history and treatment prescription, levels of blood lipids and blood glucose, liver function index and renal function index will be collected before randomisation by reviewing the medical records of the patients.

### Primary efficacy parameters
The primary efficacy parameter is the change in 24-item Hamilton Depression Rating Scale (HAMD-24) scores from the baseline at the end of the treatment. We plan to measure the response rate and remission rate as the

primary efficacy parameters, determined by changes in the HAMD-24 scores from baseline at the end of the treatment. Response to treatment is defined as a reduction of at least 50% in the HAMD-24 total score, with a final HAMD-24 score below 9. Remission is defined as a reduction in the HAMD-24 total score of at least 50% from baseline. As a valid and reliable scale for assessing the severity of depression, the HAMD scale has been widely used in studies of PSD and shows a good sensitivity in assessing treatment responses. The HAMD-24 scale consists of subscales that examine interest, anxiety, hopelessness, sleep status and self-abasement, and a higher score suggests a poorer condition of mental disorder.[17] The HAMD-24 scale will be scored three times at 0, 2 and 4 weeks after the commencement of the treatment.

### Secondary outcome measures
#### DTI results
Nerve fibres are important components of the LCSPT circuit apart from the cerebral cortex, corpus striatum and thalami. The nuclei are connected by abundant nerve fibres. DTI is a special sequence of MRI, which is currently the best non-invasive method to observe and

| Table 2 | Active rTMS treatment protocols | | | | | |
| --- | --- | --- | --- | --- | --- | --- |
| **Localisation** | **Frequency** | **Intensity** | **Times/train** | **Trains** | **Duration** | **Total times** |
| Left DLPFC | 10 Hz | 110% MT | 200 | 10 | 40 s | 20 |
| Right DLPFC | 1 Hz | 100% MT | 30 | 10 | 10 s | 20 |

DLPFC, dorsolateral prefrontal cortex; MT, motor threshold.

track the nerve tracts. We will assess the DTI parameters of the patients including FA, ADC and NFN collected on enrolment and at the end of the trial.

### Neuropsychological tests

The neuropsychological status of the patients will be examined using the following scales or questionnaires: National Institutes of Health Stroke Scale (NIHSS), Activities of Daily Living Scale (ADLs), Montreal Cognitive Assessment (MoCA), which are widely used to evaluate neurological functions; the Clinical Global Impressions scales (CGI), which is commonly used to measure the global symptom severity and treatment response for patients with mental disorders; Aphasia Battery in Chinese (ABC), which is used to estimate the linguistic functions of Chinese patients; and Social Support Revalued Scale (SSRS) and Medical Coping Modes Questionnaire score (MCMQ), which evaluate how social support influences the emotions of the patients. All the tests will be performed at 0, 2 and 4 weeks after the commencement of treatment.

### Assessment of adverse events

All adverse events (AEs) will be reported and recorded on the case report form (CRF) by investigators. The investigators will record the date of occurrence, duration and severity of the AEs, along with the treatments and consequences. Cases of serious AEs will be immediately reported to the project leader and the ethics committee, and the latter will evaluate whether the patient should receive further treatment. We will discontinue the allocated interventions if the disease shows significant worsening. Those participants who suffer harm from this trial will be provided with compensation.

### Quality control and data management

The trial protocol has been revised several times by experts in neurology, rehabilitation, medical imaging, statistics and psychology. All the investigators have participated in the training sessions to ensure that they fully understand the trial protocols and the standard procedures before the study. Inspectors will be nominated to regularly visit each research centre to monitor the implementation of the protocols and assess the compliance of the participants and investigators. The CRF data will be double-entered by statisticians to ensure the accuracy of the data. Statisticians in our team will be responsible for data management and analysis. To improve the participants' adherence to this study, we will select the participants rigorously and cover the costs of neuropsychological assessment and MRI. At the end of treatment the patients will be informed about the follow-up and a telephone call will be made in the fourth week after the end-point to remind them about follow-up.

### Statistical analysis

The quantitative data including the demographic indicators, laboratory test results, MRI results and neuropsychological test results will be expressed as numbers, means with SD or median with range or quartile range as appropriate. These data will be compared using the independent sample t-test or rank-sum test depending on whether the data are normally distributed. Qualitative data, including the response and emission rates and incidence of AEs, will be expressed as frequency or frequency rate and compared using a $\chi^2$ test between the two groups. The missing data will be described in terms of exit data, exit reason, treatment arm, frequencies and percentage of missing data in each group and the data imbalances will be evaluated by a $\chi^2$ test or Fisher's exact test. All the data will be double-entered into Epidata software and analysed using SAS software by statisticians blinded to the participants' allocation. A P value <0.05 will be considered to indicate a statistically significant difference.

### Monitoring

All personal information collected from the potential and enrolled participants will be stored in the medical record management system maintained by the Information Security Department of each hospital to ensure confidentiality. Investigators will record the research information on the CRFs and identify the CRFs with only initials and participation number rather than the participants' real identity. The research information has been detailed above, including neuropsychological tests, MRI results and baseline parameters. All the CRFs will be kept securely in a locked area. The data managers, study statisticians and investigators will have access to the final trial dataset. The data managers and study statisticians will only access the data for data management and statistical analysis and will not be involved in the clinical treatment and psychological assessment. An annual audit by the Delegation of the Research Department of Southern Medical University is provided.

## DISCUSSION

The conventional treatments of PSD include selective serotonin reuptake inhibitors,[18] tricyclic antidepressants[19] and electroconvulsive therapy (ECT),[20] which are not always effective. Recent evidence has shown that rTMS has antidepressant actions[21 22] and produces therapeutic effects comparable to those of ECT.[23] In spite of the various hypotheses that have been proposed, the exact mechanism underlying the antidepressant effect of rTMS remains unknown.[24–26]

In this trial we determined the frequency and intensity of rTMS treatment based on previous studies reporting the positive antidepressant effect of the treatment,[27–29] and chose the settings of pulses and sessions according to an evidence-based guideline.[4] As both high-frequency rTMS of the left DLPFC and low-frequency rTMS of the right DLPFC are recommended in the treatment of major depression, and considering that patients with PSD have more brain structural changes than those with primary depression, we chose bilateral DLPFC stimulation for rTMS treatment in this study.

Changes in white matter have an important role in the development of many neuropsychiatric diseases,[30] including cognitive and emotional disorders.[31] As a non-invasive imaging technique, DTI provides an important modality for evaluating white matter integrity in vivo. In DTI, the parameter FA can sensitively reflect the integrity of the white matter fibres while the ADC value allows sensitive detection of changes in the density and number of nerve fibres.[26] Numerous studies have reported reduced FA values in the brain of patients with depression,[26 32] and the structural and functional brain abnormalities, particularly in the LCSPT system, are thought to be associated with the occurrence of depression.[33] Our previous study has confirmed that the occurrence of PSD was associated with neurological functional deficits following basal ganglia infarction, and the depression level was correlated with the reductions in the FA values of the bilateral pallidum and the left putamen.

DTI offers an important means for evaluating the therapeutic effect in patients with PSD. Allendorfer et al[15] compared the DTI data collected before and after 10-day treatment with intermittent theta burst stimulation from eight patients with post-stroke aphasia and found significant increments in the FA values near the inferior and superior frontal gyri and the anterior corpus callosum after treatment. Liao et al[34] reported that, in patients with treatment-resistant depression, the FA value decreased significantly in the white matter in the left middle frontal gyrus, and 4 weeks of active rTMS treatment resulted in obvious improvement of the decreased FA values. Lyden et al[35] also confirmed the positive effect of 1 week of ECT on major depression by reporting significant increases in the FA values in the bilateral anterior cingulum and forceps minor after the treatment. Compared with major depression, PSD is associated with more severe structural and functional abnormalities of the brain, but there are currently insufficient studies of white matter integrity in patients with PSD using DTI. In this study we will compare the DTI data before and after 4 weeks of rTMS treatment as a secondary outcome parameter to assess the effect of rTMS on white matter integrity in patients with PSD.

This trial is expected to produce additional evidence to verify the effects of rTMS treatment on depression after ischaemic stroke in the basal ganglia area; we also aim to explore the role of the LCSPT circuit in PSD. Findings from this study will enrich randomised controlled trial data of rTMS for PSD in Chinese patients, and contribute to the clinical efficacy evaluation of rTMS and also to the improvement of the neural loop theory.

**Acknowledgements** The authors thank all those who participated in the trial.

**Contributors** SZ, SP, MS and LZ participated in the conception and design of the trial. AC, JZ and LZ participated in planning the analysis of the data. YT, AC and LY participated in data collection and recruitment of participants. AC, LY, SP, MS and LZ participated in the treatment of participants. YT and LZ participated in drafting the manuscript. All the authors discussed, revised and approved the final manuscript.

**Funding** This study was supported by grants from the Training Program of the Clinical Research Plan of Southern Medical University (LC2016PY031).

**Competing interests** None declared.

**Patient consent** Obtained.

**Ethics approval** The ethics committee of Zhujiang Hospital of Southern Medical University approved the study (Identifier: 2016-SJNK-005).

**Provenance and peer review** Not commissioned; externally peer reviewed.

**Data sharing statement** The results of this original research will be disseminated by publication in a peer-reviewed journal and presentation at international conferences.

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
