## [Reviewer comments · BMJ Open]

ARTICLE DETAILS

TITLE (PROVISIONAL)	Repetitive transcranial magnetic stimulation for depression after basal ganglia ischemic stroke: protocol for a multicenter, randomized, double-blind, placebo-controlled trial
AUTHORS	Tang, Ying; Chen, Aimin; Zhu, Shuzhen; Yang, Li; Zhou, Jiyuan; Pan, Suyue; Shao, Ming; Zhao, Lianxu

VERSION 1 – REVIEW

REVIEWER	Dr. Levent Sevincok, M.D., Adnan Menderes University, Department of Psychiatry Turkey
REVIEW RETURNED	27-Jul-2017

GENERAL COMMENTS	My suggestions are: 1.current or prior antidepressant use for any reason should be stated in the Methods as an inclusion or exclusion criteria.2.appropriate to the study design, only the stroke patients due to basal ganglia lesions will be included in the study, should be stated.3.The type of the stroke (Hemorrhage or infarkt) should be stated.2.Why will the left-handedness patients not be included in the study, should be stated in method, should be explained.3.Clinical Global Impression Scale should be added as an outcome criteria.4.The lesion characteristics are important variables in investigating the treatment efficacy in PSD patients. So, the lateralization, number and size of the lesion, if any, the number, and localization of previous strokes should also be recorded.5.Will any other treatments be used such antidepressants, anxiolytics, or hypnotics during the study, should be explained.
---

REVIEWER	Colleen Mills-Finnerty, Ph.D. Veterans Administration, Palo Alto, California Stanford University Dept. of Psychiatry & Behavioral Sciences
REVIEW RETURNED	14-Aug-2017

GENERAL COMMENTS	The aim of this protocol is to assess the effect of a randomized, double blind, short term rTMS treatment intervention on post-stroke depressive symptoms as well as white matter integrity. Overall the study seems appropriately designed to test the stated aims; however, the protocol would benefit from greater discussion of the rationale and limitations of this approach.
---

	Specifically, the choice of rTMS protocol requires substantial elaboration, particularly the rationale for using two different stimulation protocols to left vs. right DLPFC as well as two different intensities relative to motor threshold. The authors should discuss:  1. The motivation for this choice 2. Clear hypotheses about the mechanism of the stimulation protocol in remediating depression 3. Cite the relevant literature to demonstrate that this protocol is the most appropriate for post-stroke depression specifically. The protocol would also benefit from more discussion related to the use of DTI as a secondary outcome measure.  1. How sensitive is DTI to measure short term changes? Please cite relevant literature. 2. Will DTI be collected at one site or at multiple sites? If multiple sites, please describe how effects of site will be handled in data analysis, given the challenges of comparing images acquired from different MRI scanners. 3. The DTI data quality control and data analysis plan deserves more discussion. What quality control will be done on the images to ensure that false positives will be appropriately controlled, given that DTI values can suffer from unreliable signal due to susceptibility artifact (see Pujol et al., 2015, Wang et al., 2017, Lilja et al. 2015)? 4. Given the power estimate is based on rTMS treatment response effect size, I am concerned that the imaging component of the study will not be powered to robustly detect individual differences with $n < 30$ especially if data quality control is not planned for in advance and leads to attrition beyond the estimated 25% drop out rate. This is concerning because DTI data on such a sample would be valuable to the field and understanding individual differences would have greater explanatory value than simply characterizing differences between groups. 5. It would be helpful to state a priori hypotheses about this secondary outcome variable, e.g. define which ROIs are expected to change in response to rTMS on which measures, and test outcomes only in that ROI. Or, if the aim is exploratory, a summary of what analyses will be conducted and how false positives will be controlled for is needed.
--	---

VERSION 1 – AUTHOR RESPONSE

Responses to Reviewer 1, Dr. Levent Sevincok, M.D.

1. Comment: state current or prior antidepressant use for any reason in the Methods as an inclusion or exclusion criteria.

Response: We added “current or prior antidepressant use for any reason” in the exclusion criteria of the participants.

2. Comment: state only the stroke patients due to basal ganglia lesions will be included in the study.

Response: Thanks for reminding us of this point. We have specified in the inclusion criteria that only the patients with first-time ischemic stroke will be included in the study.

3. Comment: state the type of the stroke (Hemorrhage or infarct).

Response: We have stated in the inclusion criteria that only the patients with first-time ischemic stroke will be included in the study.

4. Comment: explain the reason of the inclusion criteria that left-handedness patients not be included in the study.

Response: There are some differences in the dominant hemisphere between left-handed and right-handed patients. In order to eliminate the potential impact of such differences on the results, we chose only right-handed patients also based on the consideration that the brain may have more severe structural damage in PSD patients and most of the Chinese are right-handed.

5. Comment: add Clinical Global Impression Scale as an outcome criteria.

Response: We have added the Clinical Global Impression Scale as one of the secondary outcome measures.

6. Comment: record the lesion characteristics.

Response: We added in the text that the characteristics of the lesions will be recorded for baseline assessment.

7. Comment: explain will any other treatments be used such as antidepressants, anxiolytics, or hypnotics during the study.

Response: We have specified in the “interventions” section in the revised manuscript that the patients will not receive any other antidepressants, anxiolytics or hypnotics treatments that may potentially affect depression treatment during the study.

Responses to Reviewer 2, Colleen Mills-Finnerty, PhD

1. Comment: elaborate the choice of rTMS stimulation protocol

Response: We have re-written the background and rationale in the Introduction section and the “discussion” section, where we discussed the reason for this choice and proposed our hypotheses concerning the mechanism of the stimulation protocol in ameliorating depression.

2. Comment: more discussion related to the use of DTI as a secondary outcome measure.

Response: According to reviewer’s suggestion, we made elaboration in the “magnetic resonance imaging” section, where we provide details of the ROIs, DTI data processing and quality control method.

Meanwhile we have also re-written the “discussion” section, where we cited relevant studies that supported the sensitivity of DTI in detecting short-term changes and discussed our hypotheses that the antidepressant effect of rTMS is related to its action on the LCSPT circuit in patients with depression following cerebral infraction in the basal ganglia area.

We tried our best to improve the manuscript and made some revisions in the manuscript based on the SPIRIT checklist without changing the primary content or the framework of the paper. Due to the number of changes we have made in the text, we did not list these changes but highlighted them in red in the revised manuscript.

We appreciate the Editors and Reviewers’ hard work and hope that this revised manuscript will be suitable for publication.

Once again, thank you very much for your comments and suggestions.

VERSION 2 – REVIEW

REVIEWER	Colleen Mills-Finnerty, Ph.D. Stanford University, Dept. Of Psychiatry and Behavioral Science Veteran's Administration Palo Alto, Mental Illness Research, Education, and Clinical Care Advanced Fellowship Program
REVIEW RETURNED	21-Oct-2017
GENERAL COMMENTS	This revised protocol for measuring response to rTMS treatment in post-stroke depression is clearer about the rationale and expected outcomes in the study, particularly the use of DTI as a secondary outcome measure and the TMS intervention parameters.

VERSION 2 – AUTHOR RESPONSE

The comments are all valuable and very helpful for revising and improving our paper and provide important guidance for our further research. We have studied the comments carefully and have made corresponding revisions.